# Improving Contrastive Learning on Imbalanced Seed Data via Open-World Sampling

**Ziyu Jiang[1], Tianlong Chen[2], Ting Chen[3], Zhangyang Wang[2]**
[1]Texas A&M University, [2]University of Texas at Austin, [3]Google Research, Brain Team
`jiangziyu@tamu.edu`, `{tianlong.chen,atlaswang}@utexas.edu`, `iamtingchen@google.com`

## Abstract

Contrastive learning approaches have achieved great success in learning visual representations with few labels of the target classes. That implies a tantalizing possibility of scaling them up beyond a curated "seed" benchmark, to incorporating more unlabeled images from the internet-scale external sources to enhance its performance. However, in practice, larger amount of unlabeled data will require more computing resources due to the bigger model size and longer training needed. Moreover, open-world unlabeled data usually follows an implicit long-tail class or attribute distribution, many of which also do not belong to the target classes. Blindly leveraging all unlabeled data hence can lead to the data imbalance as well as distraction issues. This motivates us to seek a principled approach to strategically select unlabeled data from an external source, in order to learn generalizable, balanced and diverse representations for relevant classes. In this work, we present an open-world unlabeled data sampling framework called *Model-Aware K-center* (**MAK**), which follows three simple principles: (1) *tailness*, which encourages sampling of examples from tail classes, by sorting the *empirical contrastive loss expectation* (**ECLE**) of samples over random data augmentations; (2) *proximity*, which rejects the out-of-distribution outliers that may distract training; and (3) *diversity*, which ensures diversity in the set of sampled examples. Empirically, using ImageNet-100-LT (without labels) as the seed dataset and two "noisy" external data sources, we demonstrate that MAK can consistently improve both the overall representation quality and the class balancedness of the learned features, as evaluated via linear classifier evaluation on full-shot and few-shot settings. The code is available at: `https://github.com/VITA-Group/MAK`.

## 1 Introduction

Contrastive learning has been successfully applied to learning strong visual representations in an unsupervised manner [1–5]. The promise of label-free learning makes it appealing to scale up contrastive learning with massive unannotated data, e.g., from internet-scale sources [6]. However, training with a larger amount of unlabeled data is not cheap. As shown in [1, 2], contrastive learning needs a bigger model and longer training compared to supervised counterparts. With a larger amount of data, it also requires more compute resources (for bigger model and longer training). With a limited computing budget, it is likely that the out-of-distribution data in the wild would suppress the learning of relevant features.

Moreover, different from standard benchmarks that are carefully curated and well balanced across classes (e.g., ImageNet), the data distribution in the open world are extremely diverse and always exhibits long tails [7, 8]. A few very recent works [9–11] have studied whether contrastive learning can still generalize well in those long-tail scenarios. And they found that while contrastive learning learns more balanced feature space than its supervised counterpart , it still exhibits certain vulnerability

35th Conference on Neural Information Processing Systems (NeurIPS 2021).

to the long-tailed data [11]. Therefore, blindly gathering unlabeled data in the wild may exacerbate the imbalancedness issue of the training data.

Here we consider the problem of sampling open-world unlabeled data for improving the representation learning, not just for the head classes but also for the tailed classes. To make it clear, we describe the problem setting in more detail.

**Problem Setting:** Assume that we start from a relatively small ("seed") set of unlabeled training data, where the data distribution could be highly skewed yet unspecified. We aim to retrieve an extra set, with a given sampling budget, of freely available images from some external sources (e.g., the Web), to enhance self-supervised representation learning for targeted distribution (of seed set). It is worth noting that in our setup, we do not use labels for data sampling or training (e.g. [12]), but only use labels as a way to evaluate the representation. By training on the retrieved unlabeled examples, our goal is to learn "stronger and fairer" visual representations that improve not only the overall quality but also the balancedness across various class attributes.

It is worth pointing out some unique challenges in our problem setup. 1) We may start from a skewed set, and the actual class imbalancedness is unknown to us due to label unavailability, making the most approaches handling imbalance in the (semi-)supervised setting like pseudo-labeling, re-sampling, or loss re-weighting [13–18] inapplicable. 2) Adopting a pre-trained backbone trained on imbalanced seed data with tail classes under-learned may amplify unfairness, as it is harder to retrieve images for poorly learned tail classes. 3) Widely existing irrelevant outlier samples in the open world are harder to detect given the lack of label information.

In this work, we conduct the first systematic study on leveraging additional unlabeled data from external sources when performing contrastive learning on a long-tail seed training set. We observe that, while randomly sampling more unlabeled data can effectively help overall accuracy, it shows only limited and unstable benefits to the balancedness. Thus we aim to seek a principled open-world unlabeled data sampling strategy, and it follows three principles outlined below.

The first principle we follow is *tailness*. Intuitively, we would like to sample more examples that correspond to tail classes. Since neither labels nor class predictions are are unavailable, we seek another *proxy* for *tailness*, using each sample's training loss to identify "hard samples". However, compared to similar ideas adopted in supervised learning [19], our proxy signal is weaker (only contrastive loss, and not directly compared to the class label) and much noisier due to the strong random augmentations in contrastive learning. In view of that, we propose to instead use an empirical expectation of sample loss over multiple random augmentations as the proxy. Our experiments verify that this new proxy of *empirical contrastive loss expectation* (**ECLE**) can reliably and stably spotlight more samples from tail classes.

We further complement the *tailness* principle with two other principles: *proximity* and *diversity*. Without *proximity* to examples in the seed set, the large-loss samples could simply be outliers, and training with those may hamper the model generalization on the target distribution. Thus we incorporate a feature distance regularizer between new external samples and seed training samples to reject too "far-away" samples from the former. Without diversity in the retrieved images, the benefits of the additional data can be significantly lower. We hence include another diversity-promoting term in sample selection. Eventually, our ideas could be mathematically unified into one framework called *Model-Aware $K$-center* (**MAK**) (as shown in Figure 1), which could be viewed as a principled extension of the core-set active learning [20], from supervised to unsupervised representation learning.

Our contributions are summarized below.

- *New problem and insight*: leveraging external unlabeled data could help contrastive learning on imbalanced data, but needs strategic sampling to avoid backfiring on balancedness.

- *New tail sample mining for contrastive learning*: an empirical contrastive loss expectation (ECLE) is proposed to adaptively mine under-learned samples (usually on the tail classes), while alleviating the substantial impact of random data augmentations.

- *New principled sampling framework*: our unified formulation of Model-Aware $K$-center (MAK) seamlessly integrates ECLE, Out-of-Distribution (OoD) rejection and diversity promotion. It also extends the ideas of core-set active learning [20] to unsupervised representation learning.

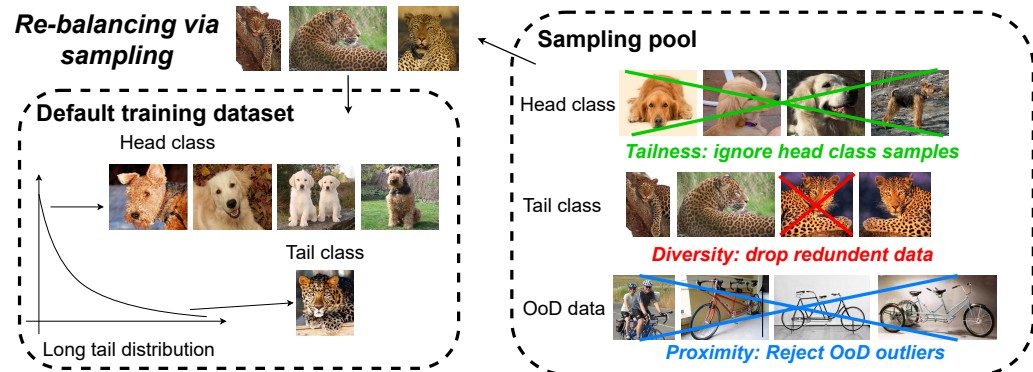

Figure 1: The overview of the proposed MAK sampling framework. MAK re-balances the long tail distribution via sampling additional data from a sampling pool. MAK are composed by three components: *tailness*, *proximity* and *diversity*.

- *Compelling empirical results*: When training on ImageNet-100-LT, the MAK can consistently yield accuracy and balancedness improvement over the competitive random sampling method across different sampling datasets and budgets.

## 2 Our Method

### 2.1 Overview

**Preliminary:** In contrastive learning, the representation is learned via enforcing an anchor sample $v_i$ to be similar to another positive sample while being different from negative samples. Among various popular contrastive learning frameworks, SimCLR [1] is easy-to-implement while yielding near state-of-the-art performance. It utilizes two augmented views of the same data as the positive pairs while all other augmented samples in the same batch are treated as negative samples. Assuming a random data augmentation function $A(\cdot, \theta)$ where $\theta$ is the hyperparameter to be sampled from some empirical distribution $\Theta$ each time. With $\theta_{i,1}, \theta_{i,2} \sim \Theta, \forall i$. The SimCLR loss associated with the $i$-th sample in the batch could be represented as:

$$\mathcal{L}_{\text{CL},i} = -\log \frac{s^\tau \left( A(v_i, \theta_{i,1}), A(v_i, \theta_{i,2}) \right)}{s^\tau \left( A(v_i, \theta_{i,1}), A(v_i, \theta_{i,2}) \right) + \sum_{v_i^- \in V^-} s^\tau \left( A(v_i, \theta_{i,1}), v_i^- \right)} \tag{1}$$

where $v_i^-$ denotes the negative samples sampled from distribution $V^-$. $s^\tau$ represents the similarity function with a temperature term of $\tau$ defined as $s^\tau(a, b) = \exp(a \cdot b / \tau)$. The overall term measures the entropy for sampling $A(v_i, \theta_{i,2})$ among all $A(v_i, \theta_{i,2})$ and $v_i^-$.

Hereinafter, we focus on SimCLR as the backbone to develop our framework. However, our idea is rather plug-and-play and can be applied with other contrastive learning frameworks adopting the two-branch design with data augmentation views [3, 4], which we will explore as our future work.

**Principles:** As demonstrated in Figure 1, the principles behind MAK are to ensure *tailness*, *proximity* and *diversity*. While *tailness* is for sampling more tail class data, *proximity* ensures the sampled data is in-distribution. Besides, *Diversity* further enforce the diversity of the sampled data, preventing sampling redundant similar samples.

### 2.2 Spotting Hard Samples from Tail Classes:A New Proxy for Tailness

In supervised learning, one can spot "hard examples" which yield the largest loss values and assign higher weights for accelerated training [19], and for handling data imbalance [21]. However, such a benefit does not extend to contrastive learning straightforwardly. Importantly, the contrastive loss (1) largely depends on the random augmentations $A(\cdot, \theta)$ and thus display high randomness. As observed in experiments, the loss variations caused by selecting two similar/dissimilar augmentation views

could outweigh the loss difference caused by sample memorization or learning difficulty. Hence naively picking samples with the largest contrastive loss (1) will only produce highly noisy selections.

To eliminate the randomness caused by augmentations, we turn to the following new proxy value for the $i$-th sample, that is designed to "smooth out" random augmentations by integrating over them:

$$\mathcal{L}^{\mathcal{E}}{}_{\text{CL,i}} = \mathbb{E}_{\theta_{i,1}, \theta_{i,2} \sim \Theta} \left( \mathcal{L}_{\text{CL,i}}(\theta_{i,1}, \theta_{i,2}; \tau, v_i, V^-) \right), \tag{2}$$

In practice, the expectation is approximated by the sample mean, e.g., drawing $\{\theta_{i,1}, \theta_{i,2}\}$ for $M$ times and then averaging corresponding $\mathcal{L}_{\text{CL,i}}$ values. We denote the new loss (2) as the *empirical contrastive loss expectation* (**ECLE**) for the $i$-th sample. Note a similar "randomized smoothing" idea was utilized before to learn robust classifiers [22, 23], though in a completely different context from ours. We then sort and choose those with the largest ECLE values (2) as hard samples. Section 3.4 experimentally verifies that ECLE eliminates the randomness within reasonable computational budgets, and larger ECLE values display correlation with *tailness*.

### 2.3 Proximity and Diversity

**Proximity:** The new ECLE proxy lays the foundation for sampling hard examples. While they are found to correlate with *Tailness*, the alignment remains to be weak and noisy in the open-world setting, since the external sources can have shifted or non-overlapped class distribution with the target ones, e.g., containing "distractor" classes. Those outliers behave like zero-shot minorities and are likely to incur large losses, since they are from completely unseen and unwanted classes. Adopting only the ECLE proxy might easily pick those outliers, hurting feature learning and generalization on the underlying distribution.

We hence construct a regularization term that promotes *proximity* via rejecting OoD outliers. Let $s^1$ be the new additional set that we have sampled from the external sources, and $s^0$ be the seed training set. We let $\Delta(x_i, x_j)$ denote the feature distance between two samples. We define $D(s^0, s^1)$ as the average feature distance between each sample in $s^1$ and its nearest sample in $s^0$, formalized as:

$$D(s^0, s^1) = \frac{1}{|s^1|} \sum_{j \in s^1} \min_{i \in s^0} \Delta(x_i, x_j) \tag{3}$$

In practice, to compute $\Delta(x_i, x_j)$, we use the normalized cosine distance, by passing $x_i, x_j$ through the currently trained backbone and using their extracted features before feeding projection heads. For further efficiency, instead of calculating $D(s^0, s^1)$ over the whole $s^0$, we pre-compute the set of feature prototypes from $s^0$ using $K$-means clustering, denoted as $s^0_p$, and then compute $D(s^0_p, s^1)$.

**Diversity:** Oversampling too many external images both would add to the training overhead, and might not necessarily help since we might get redundant samples. Assume that we have a sampling budget: $s^1 : |s^1| \leq K$, the challenge is how to attain the informative samples within the size limit. We introduce the following regularization term:

$$H(s^1 \cup s^0, S_{all}) = \max_{i \in S_{all}} \min_{j \in s^1 \cup s^0} \Delta(x_i, x_j) \tag{4}$$

Here $S_{all}$ denotes all samples from the seed training set and the external sources ($s^0, s^1 \in S_{all}$). Conceptually, minimizing the term (4) boils down to choosing $|s^1|$ center points on top of the given $|s^0|$ points, such that the largest distance between any data point from $S_{all}$ and its nearest center point from $s^1 \cup s^0$ is minimized. It equals to finding a diverse subset cover for the dataset $S_{all}$ using a minimax facility location formulation [24], also known as the name of $K$-center [20].

### 2.4 Model-Aware $K$-Center: A Unified Framework

Taking together, we end up with solving the following constrained optimization:

$$\max_{s^1 : |s^1| \leq K} \left\{ \sum_{i \in s^1} \mathcal{L}^{\mathcal{E}}{}_{\text{CL,i}} - D(s^0, s^1) - H(s^1 \cup s^0, S_{all}) \right\} \tag{5}$$

The goal of (5) is to find a sample set $s^1$ from the external source, such that it could simultaneously: (i) mine more data for tail classes while overcoming augmentation randomness, by sorting external samples by their ECLE values (*tailness*); (ii) reject the out-of-distribution outliers that might distract

---

**Algorithm 1:** A greedy heuristic to efficiently solve MAK.

---

**Require :** seed training set $s^0$, external data $S_{all} \backslash s^0$, sampling budget $K$, candidate set size $C$,
feature distance function $\Delta$ (cosine distance in practice), coefficient $\alpha \in (0, 1)$.

Train feature extractor $f$ on $s^0$ with self-supervised method;

Calculate ECLE $\mathcal{L}_{\mathrm{CL,i}}^{\mathcal{E}}$ and average feature distance $D\left(s^0, s^1\right)$ with $f$ as in Equation 2 and 3, respectively;

Summarize $\mathcal{L}_{\mathrm{CL,i}}^{\mathcal{E}}$ and $D\left(s^0, s^1\right)$ with score $q = \alpha N(\mathcal{L}_{\mathrm{CL,i}}^{\mathcal{E}}) - (1-\alpha)N(D\left(s^0, s^1\right))$ where
$N(v) = \frac{v - mean(v)}{std(v)}$ is the normalization function;

Construct set $S'$: from $S_{all} \backslash S^0$, find all samples whose score $q$ are top $C$ largest among all;

`// tailness & proximity`

Construct set $s$: Initialize $s = s^0$ ;

**while** $|s \backslash s^0| \leq K$ **do** `// Apply K-center greedy algorithm for diversity`

$\quad | \quad u = \arg\max_{i \in S'} \min_{j \in s} \Delta(x_i, x_j)$ ;

$\quad | \quad s = s \cup \{u\}$ ;

**end**

**return** $s^1 = s \backslash s^0$

---

training, by constraining feature distances from the seed set (*proximity*); and (iii) control the sample volume under $K$ while ensuring sample diversity, by $K$-center sample selection (*diversity*).

We name our optimization (5) *Model-Aware $K$-center* (**MAK**), since it inherits the vanilla "data-level" $K$-center objective (4), while injecting "model-level" sample selection criteria, including largest (smoothed) model losses as well as closer feature distances. Those "model-aware" terms are necessary in our unsupervised re-balancing goal and our open-world setting, for automatically re-balancing the additional samples in favor the seed set's minority classes, and for rejecting outliers from open-world external data, respectively.

Exactly solving (5) would be NP-hard [25], and we instead refer to an empirical greedy routine with a coordinate-descent heuristic. As indicated in Algorithm 1, we start by training a feature extractor $f$ on the seed training set $s_0$, which is then used for calculating ELCE loss $\mathcal{L}_{\mathrm{CL,i}}^{\mathcal{E}}$ and average feature distance $D\left(s^0, s^1\right)$. Afterwards, we sample a preliminary pool $S'$ with top $C$ largest tailness and proximity samples ($C > K$ is candidate set size). We employ a score $q$ to combine these two metrics, defined as:

$$q = \alpha N(\mathcal{L}_{\mathrm{CL,i}}^{\mathcal{E}}) - (1-\alpha)N(D\left(s^0, s^1\right)) \qquad (6)$$

where $N(v) = \frac{v - mean(v)}{std(v)}$ is a normalization term and $\alpha \in (0, 1)$ is a coefficient. From $S'$, we then sample a diverse subset $s^1$ of size $K$, leveraging the $K$-center greedy algorithm.

## 3 Experiment

### 3.1 Settings

**Seed Training Datasets** We employ ImageNet-100-LT as the seed training dataset, which is introduced in [11]. It is a long-tail version ImageNet-100 [26]. The sample number of each class is determined by a down-sampled Pareto distribution used for ImageNet-LT [27]. It includes 12.21K images, with the sample number per class ranging from 1280 to 5.

**Sampling Datasets** We consider two datasets for sampling: ImageNet-900 and ImageNet-Places-mix. **i) ImageNet-900** ImageNet-900 is composed by the rest part of ImageNet [28] excluding ImageNet-100 [26]. It in total contains 1.14 Million images belonging to 900 classes. **ii) ImageNet-Places-Mix** We build a sampling dataset including both in-distribution and OoD data. While ImageNet-900 is employed as source of in-distribution data, the OoD data is sampled from Places, a large scale scene-centric database. 200 random sampled classes from ImageNet-900 and Places are mixed together. The resultant dataset is called ImageNet-Places-Mix and it contains 1.24 million images in total, where 0.25 million and 0.99 million are from ImageNet-900 and Places, respectively. While the

Table 1: Compare the proposed MAK with random sampling and K-center [20] in terms of the *linear separability performance* and *few-shot performance* when fixing the sampling budget to be 10K images. The seed training dataset is ImageNet-100-LT. The sampling dataset is ImageNet-900(IN900) and ImageNet-Places-Mix(IPM). ↑ means the metric is the higher the better, and ↓ means the metric is the lower the better. We run each experiment three times and report the error bar. The best performance under each setting is marked as **bold**.

| Sampling dataset | Budget | method | Protocol | Many ↑ | Medium ↑ | Few ↑ | Std↓ (imbalancedness) | All ↑ |
|---|---|---|---|---|---|---|---|---|
| None | - | - | *linear separability* | 71.2±0.8 | 65.3±0.7 | 62.7±0.9 | 3.6±0.5 | 67.3±0.7 |
| | | | *few-shot* | 52.6±0.3 | 40.5±1.5 | 32.5±1.1 | 8.3±0.4 | 44.2±1.0 |
| IN900 | 10K | random | *linear separability* | 74.6±0.3 | 69.7±0.4 | 66.1±1.2 | 3.5±0.5 | 71.2±0.2 |
| | | | *few-shot* | 56.6±1.2 | 48.6±0.4 | 43.7±1.7 | 5.3±1.2 | 51.1±0.1 |
| | | K-center | *linear separability* | 73.6±0.3 | 68.6±0.8 | 64.5±0.9 | 3.8±0.3 | 70.0±0.4 |
| | | | *few-shot* | 55.0±0.4 | 45.8±0.3 | 39.1±1.1 | 6.5±0.6 | 48.5±0.2 |
| | | MAK | *linear separability* | **76.1±0.6** | **70.8±0.5** | **69.3±0.8** | **3.0±0.1** | **72.7±0.4** |
| | | | *few-shot* | **57.4±0.6** | **48.9±0.2** | **46.3±1.5** | 4.8±0.2 | **51.9±0.4** |
| | 20K | random | *linear separability* | 75.7±0.2 | 71.8±0.1 | 69.6±1.1 | 2.6±0.4 | 73.0±0.1 |
| | | | *few-shot* | 57.4±0.7 | 49.9±0.3 | 45.9±0.4 | 4.8±0.2 | 52.3±0.5 |
| | | MAK | *linear separability* | **78.0±0.8** | **73.4±0.6** | **72.4±0.3** | **2.4±0.3** | **75.1±0.6** |
| | | | *few-shot* | **59.0±0.9** | **52.9±0.5** | **50.0±0.4** | **3.8±0.5** | **54.9±0.4** |
| IPM | 10K | random | *linear separability* | 73.8±0.7 | 67.9±0.5 | 65.1±0.9 | 3.6±0.2 | 69.8±0.5 |
| | | | *few-shot* | 55.5±0.5 | **45.8±0.8** | 38.9±0.7 | 6.9±0.3 | 48.7±0.4 |
| | | K-center | *linear separability* | 73.0±0.6 | 67.7±0.1 | 65.4±1.5 | **3.2±0.4** | 69.5±0.4 |
| | | | *few-shot* | 54.2±0.1 | 45.6±0.4 | 38.4±0.9 | 6.5±0.3 | 48.0±0.3 |
| | | MAK | *linear separability* | **74.7±0.2** | **69.2±0.7** | **66.6±0.7** | 3.3±0.3 | **71.1±0.5** |
| | | | *few-shot* | **56.8±0.7** | 45.1±0.9 | **42.6±0.8** | 6.2±0.1 | **49.3±0.7** |

ImageNet-900 is a setting with a lot of in-distribution data, ImageNet-Places-Mix is built for testing a less friendly case where there is fewer in-distribution data in the sampling dataset.

**Training settings** We conduct the experiments with the SimCLR framework. We follow the settings of SimCLR [1] including augmentations, projection head setting, optimizer, temperature, and learning rate. We use Resnet-50 [29] for all experiments (including feature extractor $f$).

**Sampling settings** When calculating loss enforcing term. Augmentation repeat number $M$ is set as 5 (see explanation at 3.4). For K-mean clustering of *proximity*, the number of centers $K$ is set as 10. For the optimization process, the coefficient $\alpha$ is set as 0.3, and the candidate set size $C$ is set as $1.5 \times K$. As we will show in Section 3.3, the hyper-parameters choosing is not sensitive.

**Evaluation protocol** To verify the balancedness of a feature space, we adopt two evaluation protocols: *linear separability performance* and *few-shot performance* following [11]. *1) linear separability performance:* The balancedness of a feature space can be reflected with the linear separability w.r.t all classes [11]. Testing it involves three steps [10]: i) Pre-train a model $f$ with contrastive learning on the imbalanced ImageNet dataset ii) Fine-tune a linear classifier with visual representation produced with a balanced dataset (by default, the full dataset where the subset is sampled from) iii) Testing the accuracy on testing dataset for the linear classifier *2) few-shot performance:* Few-shot learning is an important application for Contrastive Learning [2]. Testing it can better reflect the influence of feature embedding for down-stream tasks. The main difference on testing *few-shot performance* compared to *linear separability performance* lies in step ii): the **whole model** are fine-tuned on **1%** samples of the full dataset from where the long tail dataset is sampled. Note that we choose to tune the whole model instead of employing linear evaluation as in [11] since it can yield higher accuracy and can be a more practical setting.

**Evaluation metrics** For measuring the balancedness, we divide dataset into three disjoint groups *Many, Medium, Few* according to the size of each class following OLTR [27]. Specifically, *Many* includes classes each with over 100 training samples, *Medium* includes classes each with 20-100 training samples and *Few* includes classes under 20 training samples. For both *linear separability performance* and *few-shot performance*, we report the average accuracy of classes for each group. The standard deviation (**Std**) of three groups' accuracy is used for measuring imbalancedness [11].

Table 2: Ablation of each component of the proposed MAK when optimizing different terms with respect to *linear separability performance* and *few-shot performance* when fixing sampling budget to be 10K images. The seed training dataset is ImageNet-100-LT. The sampling dataset is ImageNet-900. The first two rows denote the random sampling baseline. ↑ means the metric is the higher the better, and ↓ means the metric is the lower the better. We run each experiment three times and report the error bar. The best performance under each setting is marked as **bold**.

| Tailness | Proximity | Diversity | Protocol | Many ↑ | Medium ↑ | Few ↑ | Std ↓ (imbalancedness) | All ↑ |
|---|---|---|---|---|---|---|---|---|
| | | | *linear separability* | 74.6±0.3 | 69.7±0.4 | 66.1±1.2 | 3.5±0.5 | 71.2±0.2 |
| | | | *few-shot* | 56.6±1.2 | 48.6±0.4 | 43.7±1.7 | 5.3±1.2 | 51.1±0.1 |
| ✓ | | | *linear separability* | 74.5±0.6 | 69.2±0.6 | 66.3±1.1 | 3.4±0.6 | 70.9±0.4 |
| | | | *few-shot* | 55.7±0.4 | 46.5±0.5 | 40.2±1.6 | 6.4±0.4 | 49.3±0.4 |
| | ✓ | | *linear separability* | 74.0±0.9 | 68.4±0.7 | 65.5±1.3 | 3.6±0.3 | 70.2±0.4 |
| | | | *few-shot* | 55.0±0.2 | 46.6±0.3 | 40.8±1.4 | 5.8±0.5 | 49.1±0.3 |
| | | ✓ | *linear separability* | 73.6±0.3 | 68.6±0.8 | 64.5±0.9 | 3.8±0.3 | 70.0±0.4 |
| | | | *few-shot* | 55.0±0.4 | 45.8±0.3 | 39.1±1.1 | 6.5±0.6 | 48.5±0.2 |
| ✓ | ✓ | | *linear separability* | 75.8±0.4 | 69.9±0.3 | **69.8±1.3** | **2.9±0.5** | 72.2±0.2 |
| | | | *few-shot* | **57.7±0.7** | 48.0±1.0 | **46.4±0.7** | 5.0±0.7 | 51.5±0.3 |
| ✓ | ✓ | ✓ | *linear separability* | **76.1±0.6** | **70.8±0.5** | 69.3±0.8 | **3.0±0.1** | **72.7±0.4** |
| | | | *few-shot* | 57.4±0.6 | **48.9±0.2** | 46.3±1.5 | **4.8±0.2** | **51.9±0.4** |

## 3.2 MAK Yield Higher Accuracy and Balancedness Improvement

We compare three sampling methods: random, K-center, and MAK at Table 1. When the sampling dataset is ImageNet-900, and the sampling budget is set as 10K, all three sampling methods could improve accuracy compared to the baseline without sampling additional data. Among them, the proposed MAK yielded the highest performance in terms of balancedness and accuracy. Specifically, compared to the closest competitor, it increases the [Std(imbalancedness), Accuracy] by [−0.5%, 1.5%], respectively, in terms of the *linear separability performance*. Also, for the *few-shot performance*, it improves [Std(imbalancedness), Accuracy] by [−0.5%, 0.8%], respectively. It is worth noting that the proposed MAK improves the performance of *Few* groups by at least 2.6% compared to other sampling methods, indicating that MAK is beneficial for tail classes. The improvement is consistent as the sampling budget increases yp to 20K, showing the robustness of MAK towards the sampling budget. An intriguing finding is that the naive K-center algorithm fails to improve the performance compared to the random sampling baseline. The intuition behind this is that too large diversity can compromise the difficulty of negative samples, overfitting the model.

Besides, for the sampling dataset with a large amount of OoD data, the proposed MAK can still yield consistent performance and balancedness improvement. Compared to random sampling baseline, it improves [Std(imbalancedness), Accuracy] by [−0.3%, 0.3%] for *linear separability performance* and [−0.7%, 0.6%] for *few-shot performance*, respectively. Compared to the K-center, while the Std of MAK is marginally lower than K-center sampling by 0.1%, it yields higher performance in terms of all other three metrics.

## 3.3 Ablation Study

**Components ablation:** In this section, we ablation study the performance with different components in MAK. As shown in Table 2, when employing any single component of the MAK (*tailness*, *proximity* or *diversity*) the accuracy of *linear separability performance* and *few-shot performance* would decrease compared to random sampling baseline. The intuition behind is: **i) Only *tailness*:** many OoD outliers would be sampled. **ii) Only *proximity*:** it would lead to large redundancy since samples sampled are all similar to the seed training dataset. **iii) Only *diversity*:** too weak negative samples (Same as sampling with K-means). *By combine tailness and proximity, the accuracy and balancedness of both protocols could achieve an improvement over the random sampling baseline. By combining the diversity term, the accuracy can be further improved while the balancedness stays in the same level.*

**Hyper-parameter robustness:** We further look into the sensitivity of the hyper-parameters in the proposed MAK. As demonstrated in table 3, compared to default settings, when using bigger $K$s, the accuracy of *linear separability* marginally drops while the balancedness fluctuates (which may be caused by randomness). For *few-shot performance*, the accuracy keeps at the same level while the balancedness marginally drops. When it comes to $\alpha$, there is only a marginally change when $\alpha$ increases to 0.5 and 0.7 for all metrics except balancedness of *few-shot performance*, which marginally decreases. For *linear separability performance*, employing larger $C$ could lead to a marginally drop in accuracy while improving balancedness. Meanwhile, for *few-shot performance*, the accuracy improves while the balancedness fluctuates.

### 3.4 Further Analysis

**ECLE proxy correlates with data distribution** We start by verifying if the ELCE proxy can reflect the *tailness*. To this end, in the test set of ImageNet-100-LT, we measure the likelihood of sampling tail classes when sampling samples with large ECLE. For fair comparing, we employ the metric of $\phi = \frac{\text{target group's percentage in samples with 10\% highest loss}}{\text{data percentage of the target group}}$ to mitigate the influence of different group size . As demonstrated in Figure 2, as the repeat number $M$ increases, $\phi$ of $few$ and $medium$ would increase while $\phi$ of $many$ decrease. When $M = 10$, $\phi$ would become to be steady with respect to the repeat number, where $\phi$ of the minority is almost twice larger than the majority, which proves that ECLE can be used to spotlight the tail classes with a small $M$. According to the empirical result, we adopt $M = 10$ for MAK in practice. Note that $M = 10$ corresponds to 5 times of forward pass for SimCLR, which is a reasonable computational cost.

Table 3: Study the sensitivity of hyper-parameters of MAK in terms of accuracy and balancedness for *linear separability* and *few shot* when sampling 10K samples on ImageNet 900. Table (a), (b), (c) study the K-means center number $K$, the coefficient $\alpha$ and the candidate set size $C$, respectively.

(a)

| Protocol | linear separability | | few shot | |
|---|---|---|---|---|
| $K$ | Std ↓ (imbalancedness) | All ↑ | Std ↓ (imbalancedness) | All ↑ |
| 20 | 3.3 | 72.3 | 5.2 | 51.9 |
| 50 | 2.5 | 72.1 | 4.9 | 51.7 |

(b)

| Protocol | linear separability | | few shot | |
|---|---|---|---|---|
| $\alpha$ | Std ↓ (imbalancedness) | All ↑ | Std ↓ (imbalancedness) | All ↑ |
| 0.5 | 3.0 | 72.6 | 5.4 | 51.8 |
| 0.7 | 2.8 | 72.7 | 5.1 | 51.4 |

(c)

| Protocol | linear separability | | few shot | |
|---|---|---|---|---|
| $C$ | Std ↓ (imbalancedness) | All ↑ | Std ↓ (imbalancedness) | All ↑ |
| $2.0 \times K$ | 2.0 | 72.3 | 5.3 | 52.2 |
| $2.5 \times K$ | 2.7 | 71.8 | 4.1 | 52.8 |

**Computational overhead analysis.** The computational overhead for the proposed MAK mainly lies in two aspects: 1) The training on the seed dataset. 2) Computing ECLE. The overall computation time overhead is equivalent to 700 training epochs on ImageNet-100-LT with 10K addition data. For a fair comparison, we train the closest competitor (random sampling) by 700 more epochs (1700 epochs in total), the resultant [accuracy, std (imbalancedness)] becomes [71.2±0.1, 3.4±0.5] and [51.2±0.2, 5.8±0.1] for *linear separability performance* and *few-shot performance*, respectively. Compared to training by 1000 epochs, the imbalancedness for *few-shot performance* increase while other metrics stay the same, which maybe because of the over-fitting. The proposed MAK can still surpass it in terms of performance.

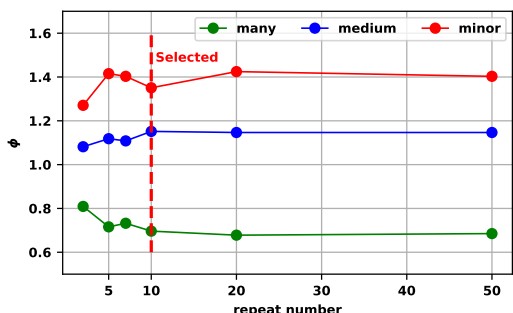

Figure 2: The correlation between ECLE and tailness on ImageNet-100-LT test dataset with respect to different repeat number $M$. The correlation is reflected with $\phi$. In practice, we select $M = 10$ as it can reflect the tailness.

**Imbalanced downstream task analysis.** To evaluate if the proposed method can address the imbalancedness for the downstream imbalance task, we linear evaluate the proposed MAK sampling

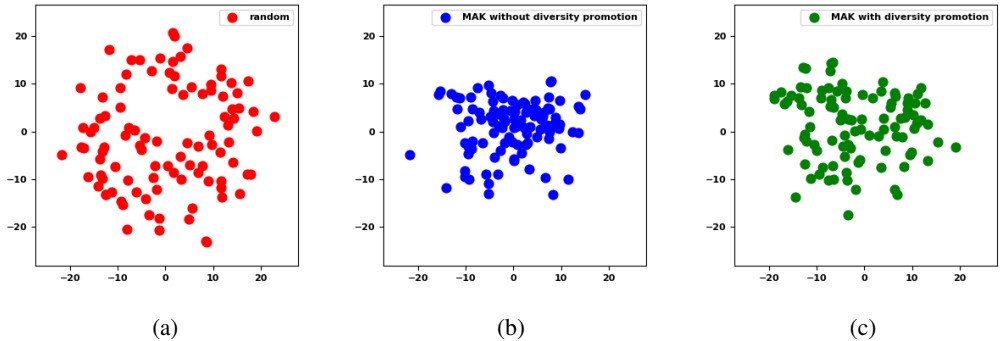

Figure 3: Compare diversity of sampled dataset among different methods with t-SNE. (a), (b) and (c) denotes the features sampled with random, MAK without diversity promotion, and MAK, respectively. Three figures are on the same scale.

method on ImageNet-100-LT. When sampling from ImageNet-900 with a 10K budget, the [accuracy, std (imbalancedness)] of the proposed MAK is [52.9±0.2, 24.8±0.2], which improves the closest competitor (random sampling) of [52.1±0.2, 25.4±0.3] via improving accuracy by 0.8 and reducing std (imbalancedness) by 0.6, demonstrating the effectiveness of our proposed method.

**Visualizing diversity.** To further verify the function of the *diversity* component, we visualize the features of the sampled dataset with different sampling method in Figure 3. Compared to the random sampling method, MAK without diversity promotion would collapse to a small area of the feature space, indicating that there are many appearance similar images. By employing diversity promotion, the features expand to large space and avoid redundancy sampling.

## 4 Related works

**Supervised Learning with Class Imbalance.** Most literature tackles the imbalance problem in supervised learning. They can be divided to two classes: class re-sampling [13–15], and loss or gradient re-weighting [16–18, 21, 30]. However, neither can extend to unsupervised setting directly, as they require either label information or label-related prediction scores. Recent works also found that training the backbone and the classifier could be decoupled[31, 32], which motivated the exploration of pre-training on imbalanced data, followed by downstream fine-tuning.

**Self-supervised Learning on Imbalanced Data.** The line of applying self-supervised learning on inherently imbalanced data stems from [9]. The authors find that simply pre-training with pre-text tasks like rotation [33] and contrastive learning (e.g., Moco [3]) could bring consistent improvements, compared to end-to-end learning without pre-training, implying its regularization effect for more balanced feature space. Follow-up works confirm the superiority of contrastive learning [10, 11] of alleviating imbalance, but also reveal that that benefit does not equal full immunity. In this work, we are the first to explore a novel direction for improving the feature balancing in contrastive learning further, via sampling additional data from the open world.

**Active learning.** Our approach is related to active learning. It has been well-studied in traditional ML literature for sampling-efficient learning, such as information theoretical methods [34], ensemble approaches [35, 36], uncertainty-based methods [37–39], and Bayesian methods[40].

In general, active learning is known to be less popular for deep learning, partially because the most effective theoretically justifiable active learning algorithms rely on finite capacity assumptions about the model class, which deep learning disobeys. Examples of recent success include core-set based [20] and loss-based training [41], both developed for supervised learning. We show that in our special setting where balancedness is of concern, sampling strategically and actively also becomes relevant.

Additionally, many have combined unsupervised learning for active learning, such as the classical pre-clustering [42] and one/few-shot learning [43]. In deep learning, pre-trained embeddings have been found to enhance active learning performances due to calibrating prediction scores better

[44–46]. Recent work [47] in turn employed active learning for efficient pre-training. None of the above concerns imbalanced issues. The closet work related to ours is [48] in (semi-)supervised deep active learning for long-tail visual recognition. The authors present feature density matching using self-supervision as well as novel pseudo-label estimators, assisted by rotation-based pre-training [49]. Differently from that, our method does not need to bootstrap from a labeled training set.

## 5 Conclusion and Discussion of Broader Impact

The data sampled from open-world always show a long tail distribution, further hurting the balancedness of contrastive learning. We tackle this important problem by proposing a unified sampling framework called MAK. It significantly boosts the balancedness and accuracy of contrastive learning via strategically sampling additional data. We believe our techniques are beneficial for improving balancedness on long-tailed data in realistic applications.

On the other hand, the proposed methods are mainly evaluated on academic datasets. When applying on real applications like autonomous driving and medical diagnosis, besides the imbalance problem, there are also problems like fair or private. This reminds us to carefully check if our method has risk of producing unfair or biased outputs in the future.

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

## Appendix: Improving Contrastive Learning on Imbalanced Seed Data via Open-World Sampling

This appendix contains the following details that we could not include in the main paper due to space restrictions.

- **(Sec. A)** Details of the computing infrastructure.
- **(Sec. B)** Details of the employed hyperparameters.

## A    Details of computing infrastructure

Our codes are based on Pytorch [50], and all models are trained with NVIDIA A100 Tensor Core GPU.

## B    Details of hyper-parameter settings

### B.1    Pre-training

We identically follow [11] for pre-training settings except the epochs number: we pre-train for 1000 epochs for all our experiments following [1] (Including the feature extractor).

### B.2    Fine-tuning

When fine-tuning for *linear separability performance*, the optimizer is set as SGD with momentum of 0.9 and initial learning rate of 30 following [51]. we train for 30 epochs and decrease the learning rate by 10 times at epochs 10 and 20. When fine-tuning for *few-shot performance*, we follow [2] fine-tuning from the first MLP projection layer. We train for 100 epochs with batch size 64. The initial lr is set as 0.02 and cosine learning rate decay without warm up is employed [2].

