# OpenReview forum: "Improving Contrastive Learning on Imbalanced Data via Open-World Sampling"
_NeurIPS.cc/2021/Conference — NeurIPS 2021 Poster_

### Official Review · Reviewer_JDg1 · 2021-06-28

**Rating:** 6
**Confidence:** 3

**Summary:**


The paper aims to seek a principled approach of selecting a subset of unlabeled data from an external source that are relevant for learning better and diverse representations. In particular, it proposes an open-world unlabeled data sampling strategy called **Model-Aware K-center** (MAK) for long-tail learning. It follows three simple principles, **tailness** to encourage sampling of examples from tail classes; **proximity** to reject the out-of-distribution outliers; **diversity** to ensure diversity in the set of sampled examples. They use ImageNet-100-LT as the target dataset and two external data sources. The experiments  demonstrate that MAK can improve the representation quality.


**Limitations And Societal Impact:**

Yes, the authors have claimed the limitations of their work.

**Main Review:**

**Strengths**
1. The paper is easy to follow. The organization is good.
2. The motivation of this work looks clear and reasonable. For open-world long-tailed unlabeled data, it is reasonable to sample more tail data, reject out-of-distribution data, and ensure diversity in the sampled set.
3. The authors demonstrate that their method improves on random selection on ImageNet-LT, and conduct ablation studies to show the separate effects of the proposed three constraints (tailness, proximity, and diversity).

**Weaknesses**
1. The proposed method lacks novelty. Firstly, for *tailness*, using the largest loss to select the hard examples is common, and using multiple random augmentations to eliminate the randomness is more like an implementation trick. For *proximity*, considering the most dissimilar samples as outliers is also common. And the *diversity* is similar to [1]. Therefore, I think the paper just applies some existing methods but lacks methodological improvements.
2. The performance gains are marginal, and the methods introduce extra computations cost compared to random selection. As shown in Table1, compared to random selection, MAK only brings about 0.9% gain on *linear separability performance* and about 0.2% gain on *few-shot performance*. In addition, MAK uses multiple random augmentations for SimCLR and a greedy heuristic algorithm, which introduces computation costs. The computation costs and training time of MAK and baseline should be added and compared.
3. From Table 2, it does look like *Diversity* does not help much. It brings no performance gain on *linear separability performance* and only 0.2% gain on *few-shot performance*.
4. The description of the Evaluation Protocol is not very clear. For *few-shot performance*, the authors claim that they use 1% samples of the full dataset from where the long tail dataset is sampled. I still can not understand how to select the 1% samples. Why not use the ImageNet-LT training set to train a classifier?

[1] Active learning for convolutional neural networks: A core-set approach, in ICLR 2018.


*****************************************************
The authors have addressed the limitation raised in the review. So I raise my rate to accept.


**Time Spent Reviewing:**

5 hours

---

> ### Author Response · Authors · 2021-08-10
> **Reply to reviewer JDg1**
>
> We thank the reviewer very much for the insightful comments and suggestions
>
> **The novelty concern**: Our novelty involves three-folds: Firstly, we provide a new formulation/setup for addressing imbalancedness of contrastive learning using sampling strategy from the free unlabeled data. The general topic of contrastive learning on imbalanced data is only explored by a handful of recent works, and further utilizing extra unlabeled in this context is unexplored by any prior work, to our best knowledge.
>
> Second, the proposed ECLE for the first time demonstrates that the few-class sample closely correlated with the unsupervised loss after eliminating the augmentation randomness. The latter point is being uniquely motivated by the status quo of unsupervised learning (relying heavily on strong augmentations) and distinct from the more direct “biggest loss” method in supervised learning. We respectfully disagree that it is an “implementation trick”, but rather a well-motivated technique grounded on unsupervised learning practice.
>
> Besides, the proposed MAK improves both performance and balancedness by seamlessly integrating tailness, proximity, and diversity. It’s worth noting that if only one element of the three is adopted, the resultant performance would fail to beat the random sampling baseline. Therefore, the organic unification of the three factors into one is the key to address our new challenging problem. We’d also like to (respectfully) point out that other reviews unanimously consider our method or problem formulation novel.
>
> **For the accuracy concern**:  The accuracy of our approach can be further improved with better hyper-parameters. For example, we found that $K=10$ in K-means of proximity (instead of 100 in the paper) further boost the result:  When sampling from ImageNet-900 with a budget of 10k, it can achieve [accuracy, std (imbalancedness)] of [72.3 ± 0.2, 3.0 ± 0.5] and [50.2 ± 0.2, 4.4 ± 0.6] for full-shot and few-shot linear evaluation performance, respectively. Compared to the closest competitor, it achieves a non-trivial improvement of [1.1, -0.5] and [1.1, -0.2] in terms of [accuracy, std (imbalancedness)] for linear separability and few-shot performance, respectively.
>
> **Add computational cost and compare**: When sampling on Imagenet-900 with a sampling budget of 10k, the additional computational cost of MAK is equivalent to 84 training epochs. For a fair comparison, we train the closest competitor (random sampling) by 100 more epochs, the resultant [accuracy, std (imbalancedness)] becomes [71.2 ± 0.1, 3.3 ± 0.5] and [49.2 ± 0.2, 4.7 ± 0.3] for linear separability and few-shot performance, respectively. In contrast, the new MAK framework can still improve [1.1, -0.3] and [1.0, -0.3] in terms of [accuracy, std (imbalancedness)] for linear separability and few-shot performance, respectively.
>
> **The function of diversity**: By comparing the last two ablation studies in Table 2, we can see that employing diversity can lead to a significant improvement on few-shot balancedness by reducing std(imbalancedness) by 0.9, demonstrating the importance of learning a more balanced representation.
>
> **How 1% samples are sampled**: The 1% samples are randomly sampled from the corresponding balanced full dataset following [1].
>
> **Using long tail dataset to train a classifier**: We follow the settings in [1] and use only balanced datasets for training classifiers to avoid introducing the prior knowledge of data imbalancedness at the evaluation stage. We think this is a more practical setting where the target data at testing comes from the unknown distribution.
>
> To further address the reviewer’s question, we tried to use the imbalanced seed dataset to train the classifier during the rebuttal. By using Imagenet-100-LT to train the classifier, when sampling on Imagenet-900 with a budget of 10k. The [accuracy, std (imbalancedness)] of the proposed MAK is [52.6+-0.4, 24.8+-0.1], which improves the the most closest competitor (random sampling) of [52.1+-0.2, 25.4+-0.3] via improving accuracy by 0.5 and reducing std (imbalancedness) by 0.6. That still demonstrates the efficiency of our proposed method.
>
> [1] Ziyu Jiang, Tianlong Chen, Bobak Mortazavi, and Zhangyang Wang. Self-damaging contrastive learning. In International Conference on Machine Learning, 2021.

---

### Official Review · Reviewer_HcS6 · 2021-06-30

**Rating:** 7
**Confidence:** 3

**Summary:**

In this paper, the author improves the accuracy and balancedness of the contrastive learning via proposing a
unified sampling framework called Model-Aware K-center, which follows the principle of tailness, proximity and diversity. The extensive experiments verified the consistent improvement compared to other sampling methods.


**Limitations And Societal Impact:**

No potential negative societal impact

**Main Review:**

In this paper, the author improves the accuracy and balancedness of the contrastive learning via proposing a
unified sampling framework called Model-Aware K-center, which follows the principle of tailness, proximity and diversity. The extensive experiments verified the consistent improvement compared to other sampling methods.

Pro:
1. the idea sampling open-world unlabeled data for improving the representation learning is interesting.
2. New problem and insight to contrastive learning.
3.The performance of experiments is  good.
4. Good writing.


Some issues:
1. the idea ( sampling open-world unlabeled data) is similar (same) to paper Learning bounds for open-set learning (ICML 2021).  Could you discusses the relationship in your paper ?

2. Contrastive learning with the idea  sampling open-world unlabeled data can be used to solve open-set learning ?

3. The  curse of dimensionality will happen ? sampling open-world unlabeled data may face the curse of dimensionality, becausw you do not know the manifold dimension of datasets. If it happens, how can you solve it ?

Please answer above issues.


**Time Spent Reviewing:**

4 hours

---

> ### Author Response · Authors · 2021-08-10
> **Reply to reviewer HcS6**
>
> We thank the reviewer very much for the insightful comments and suggestions
>
> **Relation with Learning bounds for open-set learning**: Thanks for mentioning this interesting paper. [1] addresses the open-set learning problem via constructing an ideal auxiliary domain that contains unknown classes images. The detection of unknown classes is related to the proposed proximity. However, our work is different from [1] because 1) we are addressing two different problems. Based on the best of our knowledge, we for the first time explore addressing imbalancedness for contrastive time via utilizing extra unlabeled data.  2) In addition to the outlier sampling, our work points out that improving balancedness also requires tailness and diversity.
>
> **Solving open-set learning?**: For solving the open-set problem, it is possible to employ contrastive learning to pick unknown images via sampling those images that are dissimilar from the known samples. Further, blessed for its unsupervised manner, contrastive learning is able to incorporate unknown images for training to improve unknown image picking accuracy. Besides, incorporating contrastive learning to existing frameworks such as AOSR [1] for enhancing feature encoding is also interesting.
>
> **The curse of dimensionality**: In the current setting, we avoid the curse of dimensionality by reshaping each input image to be standard 224*224, which means the output dimension would also be fixed.
>
> [1] Fang, Zhen, Jie Lu, Anjin Liu, Feng Liu, and Guangquan Zhang. "Learning Bounds for Open-Set Learning." In International Conference on Machine Learning, pp. 3122-3132. PMLR, 2021.

---

### Official Review · Reviewer_DGts · 2021-07-12

**Rating:** 7
**Confidence:** 5

**Summary:**

-This paper aims to improve contrastive learning by incorporating more open-world unlabeled data. However, such data may introduce a class imbalance problem. To solve the problem, the authors propose a sampling strategy to select a subset of the imbalanced unlabeled data following three principles: tailness, proximity, and diversity, which encourages sampling tail-class examples, rejecting outliers, and ensures diversity in the selected set. They also provide extensive experiments which demonstrate the effectiveness of the proposed sampling method.

**Limitations And Societal Impact:**

-In the introduction, the authors mentioned using each sample’s training loss to identify “hard samples”...It is unclear to me what “hardness” means in the context of contrastive learning. Is it more accurate to say under-learned samples, or rare samples? I think the word “hard sample” is more often used in supervised learning, as shown in line 113, they yield large loss values. And it is often used in label-noise learning literature, where they differentiate noisy (hard) and non-noisy data by large and small loss respectively.

-In line 103, it seems that the right-hand side of s^\tau(a,b) does not include b?

-What is the \epsilon in Eq. 2?

-I wonder if the authors have considered other distances for Eq 3, e.g., other norms? Is this an important factor in the experiment?

-How was the threshold t in Algorithm 1 selected in the experiments?

-In the experiment section, it makes sense that the balancedness may be reflected by the linear separability of all classes in the feature space, so the linear separability and the few-shot performance (accuracy) may be useful for evaluating the balancedness. But solely saying that the standard deviation (std) of the accuracy measures the imbalancedness may be a bit misleading.

-In Table 1, if we increase the sampling budget, we can see that the std of linear separability performance of the proposed method gets better. But it seems that the std of the few-shot performance gets a bit worse. Could the authors elaborate more on this inconsistency? I guess it may be some randomness. But from the current table, we may conclude that the imbalancedness does not get better in such cases, which may not be true. Because the accuracy clearly improves. I hope the authors add more explanations on this point.

--Minor typo

-Line 67: “an empirical expectation of loss” is a bit weird to me, maybe using “an empirical version of the expected loss” is better

-Line 84: adaptive-->adaptively

-Line 86: the abbreviation MAK is already introduced in Line 77

-Line 87: OoD firstly appears without the full word

-Line 100: .-->,

-Line 102: .-->,

-Line 127: varies-->verifies

-Line 187: togather-->together

-Line 210: OLTR firstly appears without any explanation

-Line 213: We-->we


**Main Review:**

-The writing of the paper is clear and easy to follow. The treatment is thorough, proceeding from introducing the SimCLR backbone, designing the sampling objective with the tailness, proximity, and diversity principles, providing a heuristic algorithm for it, to implementing experiments for the proposed method.

-The ablation study in the paper decomposes the three components of the proposed method and it is interesting to see how each of them contributes. And the effort made on demonstrating the hyperparameter robustness is also a positive point to apply the proposed method for general purposes.


**Time Spent Reviewing:**

10hours

---

> ### Author Response · Authors · 2021-08-10
> **Reply to reviewer DGts**
>
> We thank the reviewer very much for the insightful comments and suggestions
>
> **About hard samples**: Following the supervised setting, We use “hard sample” to refer to those samples whose loss expectation is high. We agree that “under-learned samples” can be appropriate.
>
> **About $s^{\tau}(a,b)$**: Thanks for pointing out, this is a typo, the right version should be $s^{\tau}(a,b)=\exp (a \cdot b)/\tau$.
>
> **epsilon in equation (2)**: We use  $\epsilon$ to distinguish **expectation** contrastive loss $\mathcal{L}^\epsilon_{\mathrm{CL}, \mathrm{i}}$ from traditional contrastive loss $\mathcal{L}_{\mathrm{CL}, \mathrm{i}}$.
>
> **Other distances for Eq 3**: We didn’t consider other distances due to time constraints. This is an interesting suggestion and we would explore this in the future.
>
> **The selection of $t$**: We choose $t$ to make the number of samples being filtered by the diversity condition is 0.5-3 times of the sampling budget.
>
> **About methods to measure balancedness**: We follow the [1] to measure the imbalancedness.
>
> **Explain why imbalancedness does not improve**: Though accuracy significantly improves, this does not necessarily mean the imbalancedness can improve: As demonstrated in Table 1, when increasing sampling budget, the accuracy of few-shot performance for Many, Medium, and Few uniformly improved by 2~3, which would NOT affect the std. The std would only improve when the accuracy improvement for Few is more significant than Many.
>
> [1] Ziyu Jiang, Tianlong Chen, Bobak Mortazavi, and Zhangyang Wang. Self-damaging contrastive learning. In International Conference on Machine Learning, 2021.

---

### Official Review · Reviewer_j3dN · 2021-07-13

**Rating:** 7
**Confidence:** 5

**Summary:**

This paper improves the accuracy and balancedness of contrastive learning by proposing a unified sampling framework called Model-Aware K-center. It follows three principles: tailness, proximity, and diversity. As a new problem setting, the authors reported extensive experiments to show their method's consistent improvement over other sampling methods.

**Ethical Concerns:**

No ethical concern was found.

**Limitations And Societal Impact:**

The authors adequately addressed the limitations and potential negative societal impact of their work.

**Main Review:**

Pros:
+ This paper studies an important new problem. The data sampled from the open world always show a long tail distribution, thus potentially hurting the balancedness of contrastive learning and leading to “unfair” downstream class transferability. This is a refreshing yet timely angle to study contrastive pre-training.
+ The authors also clearly lay out the unique challenges in this new setting, which is helpful to understand. The three principles and the optimization formulation look novel to me. The finding of ELCE (the big influence of augmentation randomness) is intuitive but sensible.
+ Since this is a new problem, I especially like that the authors perform due diligence in designing benchmark experiments. They created two mixed unlabeled sets for sampling that purposely include OoD samples at different levels. ImageNet-LT is adopted as the main testbed. The reported numbers and ablation studies are thorough and convincing.
+ The writing quality is very high. The paper is clearly structured, and the logic is coherent. The mathematics is concise and correct to my best knowledge.

Cons:
-  While I understand this paper's main point is on improving balancedness, I have to point out the accuracy gain looks underwhelming.  Especially, while linear separability has an okay 0.9% margin over the closest competitor, the few-shot gain is much less impressive (only 0.2%). Since few-shot is often the main application scenario, this brings up a concern for MAK's practicality.
- I’m not sure if K-center is necessary. Why not more computationally convenient alternatives, such as K-means? Also, correct me if I’m wrong -- the actual Algorithm 1 does not involve solving K-center, but instead used a check-and-go heuristic to inject diversity. Am I understanding that right?
- Computing ECLE could be expensive due to multiple feedforward passes.

**Time Spent Reviewing:**

4

---

> ### Author Response · Authors · 2021-08-10
> **Reply to reviewer j3dN**
>
> We thank the reviewer very much for the insightful comments and suggestions
>
> **For the accuracy concern** : The accuracy of our approach can be further improved with better hyper-parameters. For example, we found that $K=10$ in K-means of proximity (instead of 100 in the paper) further boost the result: When sampling from ImageNet-900 with a budget of 10k, it can achieve [accuracy, std (imbalancedness)] of [72.3 ± 0.2, 3.0 ± 0.5] and [50.2 ± 0.2, 4.4 ± 0.6] for full-shot and few-shot linear evaluation performance, respectively. Compared to the closest competitor, it achieves a non-trivial improvement of [1.1, -0.5] and [1.1, -0.2] in terms of [accuracy, std (imbalancedness)] for linear separability and few-shot performance, respectively.
>
> **Can we use K-means**: While sampling, each image corresponds to one feature in the feature set. Given K-means may NOT give features from the feature set, we cannot link such features back to images. In contrast, K-center is a discrete version of K-means, ensuring the representative feature comes from the given feature pool.
>
> **Does Algorithm 1 involve solving K-center**: We simplify the greedy K-center algorithm in the proposed algorithm: we keep the greedy K-center algorithm’s metric for measuring the distance of an image to a dataset. Meanwhile, instead of picking the “farest” samples every iteration, we simplify it by introducing a threshold to pick sufficiently “far” samples.
>
> **ECLE cost concern**: We conduct an ablation experiment by controlling the computational cost to be the same: When sampling on Imagenet-900 with a sampling budget of 10k, the additional computational cost of MAK is equivalent to 84 training epochs. For a fair comparison, we train the closest competitor (random sampling) by 100 more epochs, the resultant [accuracy, std (imbalancedness)] becomes [71.2 ± 0.1, 3.3 ± 0.5] and [49.2 ± 0.2, 4.7 ± 0.3] for linear separability and few-shot performance, respectively. In contrast, the MAK framework with $K=10$ can still improve [1.1, -0.3] and [1.0, -0.3] in terms of [accuracy, std (imbalancedness)] for linear separability and few-shot performance, respectively.

---

### Decision · Program_Chairs · 2021-09-27

**Decision:**

Accept (Poster)

**Comment:**

The paper worked on contrastive learning on class-imbalanced data and proposed *model-aware K-center* for the purpose. It makes use of additional unlabeled dataset which is bigger than the targeted unlabeled dataset for contrastive learning in terms of the number of instances and the number of classes. According to three principles, namely *tailness*, *proximity*, and *diversity*, certain unlabeled data are sampled from the pool dataset to re-balance the targeted dataset. The writing is clear, the motivation is strong, the idea is novel, and the results are significant. Thus, it should be accepted for publication.

It is quite interesting that the proposed method can satisfy the tailness and the proximity at the same time as shown in Figure 1, because the former sounds like accepting long-tail **classes** and the latter sounds like rejecting long-tail **instances**. Since contrastive learning should be regarded as **pre-training**, it may not be very natural if the label space for the downstream tasks is fixed at the time of contrastive learning (it is unsupervised and there is no label at all). Moreover, since the targeted dataset is smaller, is it possible that a class exist in both datasets but **no data has been drawn from this class in the targeted dataset** and then all the data of this class in the bigger pool dataset become out-of-distribution data by the proposed method (i.e, some data should be accepted but will be rejected since this class is missing in the targeted dataset)? Perhaps I have some misunderstanding because I didn't carefully go through the full paper by myself, but I believe clarifying my questions (not concerns, just questions) are very helpful and can maximize the impact of your work.

BTW, the following paper should be related to your work, which can sample in-distribution data and reject out-of-distribution data though it considered to enlarge but not re-balance/enrich the targeted dataset (i.e., quantity vs quality):

Yixing Xu, Yunhe Wang, Hanting Chen, Kai Han, Chunjing Xu, Dacheng Tao, and Chang Xu. Positive-Unlabeled Compression on the Cloud. NeurIPS 2019.